# Amateur Athlete with Sinus Arrest and Severe Bradycardia Diagnosed through a Heart Rate Monitor: A Six-Year Observation—The Necessity of Shared Decision-Making in Heart Rhythm Therapy Management

**DOI:** 10.3390/ijerph191610367

**Published:** 2022-08-19

**Authors:** Robert Gajda, Beat Knechtle, Anita Gębska-Kuczerowska, Jacek Gajda, Sebastian Stec, Michalina Krych, Magdalena Kwaśniewska, Wojciech Drygas

**Affiliations:** 1Center for Sports Cardiology, Gajda-Med Medical Center, ul. Piotra Skargi 23/29, 06-100 Pułtusk, Poland; 2Department of Kinesiology and Health Prevention, Jan Dlugosz University, 42-200 Czestochowa, Poland; 3Institute of Primary Care, University of Zurich, 8091 Zurich, Switzerland; 4Medbase St. Gallen Am Vadianplatz, 9000 St. Gallen, Switzerland; 5Faculty of Medicine, Collegium Medicum, Cardinal Stefan Wyszyński University, Kazimierza Wóycickiego 1/3, 01-938 Warsaw, Poland; 6Division of Electrophysiology, Cardioneuroablation, Catheter Ablation and Cardiac Stimulation, Subcarpathian Center for Cardiovascular Intervention, 38-500 Sanok, Poland; 7Department of Congenital Heart Diseases, The Cardinal Stefan Wyszyński National Institute of Cardiology, ul. Alpejska 42, 04-628 Warsaw, Poland; 8Department of Preventive Medicine, Faculty of Health, Medical University of Lodz, ul. Lucjana Żeligowskiego 7/9, 90-752 Łódź, Poland

**Keywords:** bradyarrhythmia, block S-A, heart rate monitors, heart rate variability, leisure time activity, athlete’s heart, pacing therapy, shared decision-making, cardioneuroablation, deconditioning

## Abstract

Heart rate monitors (HRMs) are used by millions of athletes worldwide to monitor exercise intensity and heart rate (HR) during training. This case report presents a 34-year-old male amateur soccer player with severe bradycardia who accidentally identified numerous pauses of over 4 s (maximum length: 7.3 s) during sleep on his own HRM with a heart rate variability (HRV) function. Simultaneous HRM and Holter ECG recordings were performed in an outpatient clinic, finding consistent 6.3 s sinus arrests (SA) with bradycardia of 33 beats/min. During the patient’s hospitalization for a transient ischemic attack, the longest pauses on the Holter ECG were recorded, and he was suggested to undergo pacemaker implantation. He then reduced the volume/intensity of exercise for 4 years. Afterward, he spent 2 years without any regular training due to depression. After these 6 years, another Holter ECG test was performed in our center, not confirming the aforementioned disturbances and showing a tendency to tachycardia. The significant SA was resolved after a period of detraining. The case indicates that considering invasive therapy was unreasonable, and patient-centered care and shared decision-making play a key role in cardiac pacing therapy. In addition, some sports HRM with an HRV function can help diagnose bradyarrhythmia, both in professional and amateur athletes.

## 1. Introduction

Although bradycardia and sinus node dysfunction (SND) are common electrocardiographic findings in competitive athletes, this phenomenon has not been widely documented in the scientific literature among amateur athletes [1]. SND is defined as the inability of the heart’s natural pacemaker, the sinus node, to create a heart rate (HR) that is appropriate for the body’s needs, causing irregular heart rhythms, namely, arrhythmias. SND is also known as sick sinus syndrome or sinus node disease [2].

Significant bradyarrhythmia and tachyarrhythmia may occur in active endurance athletes, although it remains controversial whether these arrhythmias persist after cessation of competitive endurance training [3]. However, if bradyarrhythmia is found in an amateur athlete whose training loads differ significantly from those of professionals (who have significantly higher loads), it may be that complete training stoppage does not have a significant therapeutic effect on rhythm and conduction disturbances.

Physical training results in bodily adaptations, especially in the organs involved in exercise [4,5,6].

In endurance sports (e.g., long-distance running or soccer), the cardiovascular system undergoes significant adaptations, including the heart, which then starts being referred to as the “athlete’s heart” [7,8]. Among many other features, the athlete’s heart is characterized by bradycardia and bradyarrhythmia [9]. While nocturnal bradycardia above 30 beats/min or nocturnal pauses of up to 4 s are commonly observed [10], nocturnal pauses in the form of sinus arrest (SA) of more than 6 s and ambiguous clinical symptoms are a significant diagnostic and therapeutic problem [1].

Furthermore, research showed that endurance training, even at professional levels, is not guaranteed to lead to the adaptations that are collectively referred to as the “athlete’s heart” [11]. Meanwhile, the common outcome of recreational/amateur activities to maintain the person’s heart as a “normal heart,” do not lead to the development of the characteristics of an athlete’s heart [12]. It is possible to objectively assess the degree of bradyarrhythmia (i.e., one of the features of the athlete’s heart) by using electrocardiogram (ECG) tests. Whenever it is feasible for the person to be subjected to 1–3 days of observation, the “gold standard” is to use the Holter ECG test [13]. It is rather common for athletes to make HR assessments during exercise; specifically, HR control during training using HR monitors (HRMs) helps them to carry out their training precisely within a predetermined load range [14].

In this case study, an amateur soccer player accidentally discovered, during consecutive nights of sleep using an HRM (Polar V800, which was used both for training and sleep) with the HR variability (HRV) function enabled, numerous pauses over 4 s, with some over 6.0 s (the longest 7.3 s). Using the Holter ECG test simultaneously with HRM, the longest pause (6.3 s) was recorded. At the same time, the readings from both Holter ECG and HRM were consistent.

A single episode of transient ischemic attack (TIA), which manifested after the athlete woke up, served as a reason for a neurologist and a cardiologist to propose, in a joint consultation with the patient, a pacemaker implantation. The suspected cause of the TIA was SA. The patient was not subjected to an ECG Holter or HRM at the time of the TIA diagnosis. Despite the recommendations by the physicians, the patient did not consent to the procedure and stopped the self-observation behavior (i.e., using the HRM). Then, 6 years after hospitalization, including 2 years of discontinued training due to depression, the patient underwent another cardiological examination showing the disappearance of bradyarrhythmia and even a tendency toward tachycardia. Accordingly, this case study aims to:note the “additional” role of some contemporary sports HRMs as reliable and relevant medical devices in the diagnosis of cardiac arrhythmias, in this case, bradyarrhythmias.draw attention to the fact of excessive bradyarrhythmias in the amateur athlete, mainly described in professional endurance athletes.describe the importance of shared decision-making in cardiac electrophysiology procedures and arrhythmia management (i.e., pacing, cardioneuroablation, observation, and deconditioning) while treating SA in both professional and amateur athletes.

## 2. Materials and Methods

### 2.1. Sports Biography

In this case study, we evaluated a 34-year-old male athlete. On the day of the first examinations, the athlete had a body height of 1.68 m, a mass of 63 kg, and a body mass index of 22.3 kg/m^2^. Six years later, when the athlete was 40 years old, these values were 1.68 m, 70 kg, and 24.8 kg/m^2^, respectively. The amateur athlete practiced various sports disciplines, mainly soccer and taekwondo, from the age of 9 to 38 years. He was an amateur soccer player for many years, specifically up to 34 years of age. Nonetheless, as he was diagnosed with SA, he stopped playing soccer and continued to practice taekwondo, albeit at a much lower intensity level than 4 years before. In the 2 years prior to the last observation (i.e., from 38 to 40 years of age), due to unspecified disease symptoms that arose during the course of the diagnosed depression, he stopped all physical activity until the control tests were finalized. A detailed description of the amount of time that the athlete used to spend on sports training is provided in Table 1.

### 2.2. Study Protocol

We analyzed a total of 10 one-hour sessions of simultaneous HRM and Holter ECG recordings during the nighttime hours to assess the accuracy of the HRM readings using the HRV function. Specifically, we evaluated, “beat-to-beat” intervals on the HRM relative to the R-R intervals indicated by the Holter ECG. Furthermore, we analyzed approximately 31,500 beats (R-R intervals). The average rhythm and the longest and shortest R-R intervals for each hour were evaluated. An example of a 30 s HRV HRM printout that was evaluated and compared with the Holter ECG recordings is shown in Figure 1. For analysis, only those fragments of the one-hour recordings were selected that did not show any reason for concerns about recording quality (without artifacts) and in which there were pauses longer than 4 s.

We performed the following examinations on the study participant during two periods: in the year of diagnosis of SA (2016) and 6 years later (2022; after 2 years of cessation of training) with Holter ECG, exercise stress test, regular ECG, and transthoracic echocardiography. We performed a TILT test once (2022).

For reliable medical records, we used the patient’s hospital discharge card as the source of the results of various medical examinations (including a head MRI) and medical history.

#### 2.2.1. ECG Tests

A standard 12-lead ECG was performed using a BTL Flexi 12 ECG device (BTL Industries Ltd., Hertford, UK).

#### 2.2.2. Transthoracic Echocardiography

The patient underwent a complete transthoracic echocardiographic examination using a GE Medical System Vivid 7 with a 2.5-MHz transducer (GE Medical Systems Information Technologies, Inc., Wauwatosa, WI, USA). M-mode, two-dimensional imaging, and Doppler techniques were used. The left ventricular end-systolic and end-diastolic volumes and the interventricular septum and posterior diastolic wall thickness diameters were measured.

The left ventricular systolic function was evaluated using left ventricular ejection fraction and global longitudinal strain. The left ventricular diastolic function was evaluated using mitral inflow velocities and tissue Doppler imaging values. Furthermore, the transmitral early diastolic (E-wave) velocity and atrial (A-wave) velocity were measured, and the E/A wave ratio was calculated. Early diastolic velocity (e’) was measured in addition to E/e’ ratio.

The right ventricular end-diastolic diameter from the parasternal long-axis view and the tricuspid lateral annular systolic velocity wave were measured using tissue Doppler imaging. The left atrial volume index was calculated using body surface area.

#### 2.2.3. Holter ECG

A 24-h Holter ECG monitoring was performed with the Holter ECG Lifecard CF apparatus and software version: Cardionavigator Plus Impresario 3.07.0158. (Reynolds Medical, Bath, UK, Fallbrook, CA, USA)

#### 2.2.4. Exercise Stress Test

A treadmill exercise stress test was performed according to the Bruce protocol using the following setup: BTL Flexi 12 ECG (BTL Industries Ltd.); treadmill BTL-770M (BTL Industries Ltd.); software BTL CardioPoint, version 6.1.7601.24545 (BTL Industries Ltd.).

#### 2.2.5. HRM

HRV measurements were obtained using the HRV function of the Polar V800 (POLAR Electro, Kempele, Finland) HRM. The HRM recordings were tested and compared with Holter ECG recordings. The results were analyzed by a highly experienced cardiologist.

#### 2.2.6. TILT Test

The TILT test was performed using the following setup: a Task Force Monitor measuring system s/n 003040i-2013-151-005-GG-0000 type 3040i; an Upright TILT table for Task Force apparatus s/n 1,051,833 type 900/90; a monitor for Task Force kit s/n YV3V025662 type E19-6 LED; a computer for Task Force s/n YLLQ033365 type Esprimo C710; and a Xerox Phaser printer for the Task Force camera. The results were analyzed by a highly experienced cardiologist.

### 2.3. Statistical Analysis

The normal distribution of compared variables was confirmed (in every case) by the results of Shapiro–Wilk tests. Statistical analyses were performed using paired *t* tests. The significance level was established at *p* < 0.05. All statistical analyses were performed using STATISTICA, version 13 (StatSoft, Tulsa, OK, USA).

### 2.4. Ethical Approval

This case report was approved by the ethical review board of the Bioethics Committee of the Healthy Life Style Foundation in Pułtusk (EC 3/2016/medicine/sports, approval date: 9 June 2016). The athlete provided his written informed consent to participate in the study and for his data to be published.

## 3. Results

### 3.1. HRM Data Analysis

A summary of the analyses of 10 one-hour sessions, during which the HRM and Holter ECG made recordings simultaneously, is presented in Table 2.

Both devices showed convergence of the assessed values, with the longest recorded pause being of 6.3 s. There were no statistical differences between the Holter ECG and HRM recordings for any of the assessed parameters: number of beats (*p* = 0.5248); mean bpm (*p* = 0.5225); minimum bpm (*p* = 0.7128); maximum bpm (*p* = 0.5312); and pause (*p* = 0.6508).

The recordings from both devices showing the longest pause (6.3 s) are shown in Figure 2.

The consistency of the recordings of the HRV function of the HRM (Polar V800) in relation to Holter ECG recordings allowed us to consider the maximum R-R interval of 7.3 s, which was recorded only on the HRM, as being a reliable finding (Figure 3).

### 3.2. ECG Tests

During the rest ECGs, we observed: a sinus rhythm of 60 beats/min and a normal ECG recording in 2016 and a sinus rhythm of 66 beats/min and a normal ECG recording in 2022 (Figure 4).

### 3.3. Echocardiography

All the evaluated echocardiographic parameters of the study participant remained within normal ranges, as shown in Table 3.

### 3.4. Exercise Stress Tests

In 2016, the exercise stress test was performed on a moving treadmill according to the Bruce protocol. The effort was discontinued due to reaching the 100% HR limit for the participant’s age. A 13.4 MET load was achieved, and the predicted load was 12.9 MET. The study participant showed age-appropriate physical capacity, no chest pain, no ST changes, normal blood pressure response to exercise, and no cardiac arrhythmias. The conclusion: a negative test.

In 2022, the same stress test was performed following the same protocol. The effort was again discontinued due to reaching the 100% HR limit for the participant’s age. A 10.2 MET load was achieved, and the predicted load was 12.0 MET. The participant showed a physical capacity that decreased slightly compared with the expectations for his age, no chest pain, no ST changes, normal blood pressure response to exercise, and no cardiac arrhythmias.

The conclusion: a negative test.

### 3.5. TILT Tests

A passive (unmedicated) TILT test (FINAPRES*, Enschede, The Netherlands) provided negative results. It revealed a normal response of blood pressure and HR to pionization.

### 3.6. Holter ECG

A series of Holter ECG examinations were performed in 2016 and again in 2022. The longest sinus pauses (6.3 s) and the most significant nocturnal bradycardia (33 beats/min) were recorded in August 2016. No other rhythm or conduction disturbances were observed again (only single ventricular and supraventricular beats). No ST-segment changes were observed.

## 4. Discussion

### 4.1. Major Findings

An amateur soccer player, by using a sports HRM for daily evaluation of exercise intensity and self-monitoring (both during and outside of physical activity), accidentally detected pauses in HR during the nighttime hours that repeatedly exceeded 4 s (maximum pause: 7.3 s; Figure 3). The Holter ECG performed simultaneously with HRM with an HRV function confirmed the accuracy of the HRM readings in both HR and pause length (Table 2). The longest pause recorded during the parallel study was 6.3 s, with a bradycardia of 33 beats/min (Figure 2). An important fact was the absence of cardiac arrhythmias other than the described SA and nocturnal bradycardia. There was no basis for the diagnosis of SND with tachycardia–bradycardia syndrome followed by significant SA. No atrial fibrillation or significant supraventricular or ventricular arrhythmias were observed.

During hospitalization in the neurological ward after TIA diagnosis, while recording the longest pauses on the Holter ECG, the patient was suggested to undergo electrophysiological treatment with the implantation of a pacemaker. Nonetheless, the study participant did not agree with the proposed treatment. After leaving the hospital, he discontinued medical control and self-monitoring behaviors and chose to perform a different type of sports training at much lower loads (taekwondo). At that time, he reported no symptoms. Two years prior to the follow-up, he stopped training due to problems related to a coexisting depression. During this time, he was taking antidepressants periodically. Then, 6 years after the diagnosis of nocturnal SA due to non-specific clinical symptoms, a series of repeated cardiological examinations (ECG, Holter ECG, TILT, and echocardiography) showed normal results for the study participant. The Holter ECG results showed not only a complete cessation of the pauses but also a tendency to tachycardia, both during the day and at night.

### 4.2. Recommendations for Treating SND with Cardiac Pacing

SND is characterized by the inability of the sinoatrial node to produce an adequate HR that meets the physiologic needs of an individual [15]. The guidelines for treating SND have not changed significantly over the years. The guidelines which were active during the period in which the amateur athlete studied by us suffered the longest pauses (in 2016; pause of 7.3 s) had been in force since 2013, and were as follows: for cases of syncope in patients with documented SA of more than 6 s, the indication is for a class IIa pacemaker implantation. In 2022 (the follow-up of the studied athlete), the newer 2021 guidelines were already in effect, in which the indications for the same symptoms were in class IIb (Table 4) [16].

Current guidelines indicate that cardiac pacing may be considered for SND when symptoms are likely to be due to bradyarrhythmia and the evidence is not conclusive (class IIb C). Nonetheless, in patients with bradyarrhythmia related to SND that is asymptomatic or due to transient causes that can be corrected and prevented, pacing is not recommended (class III C) [16].

In the case studied, the “effect of deconditioning”, which should be regarded as a “transient cause”, indicated that there was no need for cardiac pacing (class III C). Furthermore, the study participant was diagnosed with TIA in the neurological department of the hospital, although a clear cause was not found (i.e., the MRI and CT scans of the head showed normal results). These results make the validity of invasive therapies such as cardiac pacing all the more questionable.

There are many causes of sinus bradycardia and SND, which can be divided into externally and internally derived. Figure 5 shows the main causes of bradycardia and SND [16].

Physical activity being only one of the many causes of sinus bradycardia or SND does not exclude the possibility of physical activity coinciding with other causes. During the hospitalization of the study participant in the cardiology department, numerous diagnostic tests were performed in an attempt to exclude other potential causes.

### 4.3. Bradycardia or SND in Trained Athletes: The Real Reason

Sinus bradycardia, defined as a sinus rate of <60 beats/min, is common among athletes [17]. It is generally attributed to enhanced vagal tone caused by conditioning, as high vagal tone reduces HR and is therefore considered a physiological situation. Occasionally, HRs can be as slow as 30–40 beats/min at rest and decrease to <30 beats/min during sleep in highly conditioned athletes. SND can be defined as encompassing a wide spectrum of sinoatrial dysfunctions, ranging from sinus bradycardia, sinoatrial block, and SA to bradycardia–tachycardia syndrome [18,19].

Research shows that profound bradyarrhythmia in older adult athletes because of SND may increase the risk of sudden cardiac death [20]. However, despite this widespread belief, efferent vagal nerve activity to the heart’s pacemaker (the sinus node) has never been recorded. In athletes, HRV tends to be higher, and this is often regarded as evidence of their high vagal tone, which is then assumed to be responsible for bradycardia [21]. However, a causative link between autonomic nerve activity and HRV has never been demonstrated. HRV is defined as a beat-to-beat variability in HRs and is assumed to be the result of stochastic fluctuations in autonomic nerve activity to the sinus node, and changes in HRV are assumed to represent changes in the sinus node [22]. D’Souza et al. have analyzed all published data concerning the increase in HRV in athletes that could be explained by bradycardia [23]. These authors showed that to scrutinize and identify the mechanisms underlying the low resting HR in athletes, investigators blocked the autonomic nerve activity in the heart by injection of blockers (frequently but not exclusively atropine and propranolol). After reviewing these studies, D’Souza et al. remarked that no study had evidence indicating that there was complete autonomic blockade and the bradycardia in athletes was abolished. According to Boyett, if bradycardia was the result of high vagal tone, it should not be present after blocking the vagal activity to the sinus node [24]. Nonetheless, Katona et al. showed that bradycardia was larger after complete autonomic blockade [25]. Then, if bradycardia is not the result of high vagal tone in trained athletes, what would be the cause of this bradycardia?

In recent years, it has been demonstrated that dysfunctions or changes in the sinus node (e.g., in familial bradycardia) occur due to the remodeling of ion channels and related molecules in the sinus node. These remarks are somewhat unsurprising because the sinus node has an electrical function; therefore, it depends on the expression of these molecules and channels [26]. The most common reported cause of bradycardia in these various conditions is a downregulation of the funny channel (HCN4) and the corresponding funny current, an important pacemaker mechanism [27].

The arguments raised above prompted D’Souza et al., to hypothesize that bradycardia in athletes is the result of ion channel remodeling in the sinus node. Analysis of tissue biopsies from the sinus node of trained animals by quantitative PCR showed a widespread remodeling of ion channels and related molecules in the sinus node, including a downregulation of HCN4. Conclusions based on studies conducted with rats and mice suggest that the resting bradycardia in athletes is the result of a downregulation of funny channels (HCN4) and funny current. Within the sinus node of trained animals, D’Souza et al. observed a downregulation of the transcription factor Tbx3, upregulation of the neuron-restrictive silencer factor (NRSF), and a micro-RNA (miR-1). These changes are appropriate to explain the downregulation of HCN4 [23]. Although the cited studies were performed in animals, the same mechanism cannot be excluded in humans, including amateur and professional athletes.

### 4.4. Eligibility Criteria for Competitive Athletes with Sinus Bradycardia or SND

The diagnosis of sinus bradycardia in athletes requires professionals to carefully assess the patient’s history and related symptoms. For example, asymptomatic sinus pauses or SA (<3 s) are not considered clinically significant for the diagnosis of sinus bradycardia unless they are accompanied by other symptoms. Specifically, long duration pauses may fall within the spectrum of physiological responses to athletic conditioning; however, they are considered abnormal when sinus bradycardia, sinoatrial exit block, and SND with pauses at the termination of a supraventricular tachycardia (SVT) are accompanied by other symptoms. Athletes with symptoms potentially associated with these arrhythmias should be subjected to an ECG, 24 h ambulatory monitoring, and an exercise test. In symptomatic patients or those with resting HRs < 30 bpm or pauses > 3 s, clinical assessment for structural heart diseases, noninvasive assessment of sinus node function with ambulatory monitoring, and stress testing are also appropriate. However, invasive electrophysiology studies play a very limited role in the assessment of sinus node function.

Generally, research suggests that athletes with symptoms related to sinus bradycardia caused by high vagal tone related to training should restrict their athletic training and undergo clinical reassessment of symptoms and sinus node function [17]. Athletes with symptomatic bradycardia not responsive to measures such as deconditioning or the withholding of nonessential medications that are contributing to the bradycardia may need to be treated with a permanent pacemaker, although this is very rarely the case for athletes [28,29]. In summary, the recommendations for competitive athletes with sinus bradycardia or SND are as follows [17]:Athletes with asymptomatic sinus bradycardia, exit block, pauses, and arrhythmia can participate in all competitive athletic activities unless they present other underlying structural heart diseases or arrhythmias (Class I; Level of Evidence C).Athletes with symptomatic sinus bradycardia should be evaluated for structural heart disease and receive treatment for the bradycardia, generally by pacemaker implantation. They should be restricted from both training and athletic competition while under evaluation. If the bradycardia treatment eliminates the symptoms, they can participate in athletic training and competition unless they present other underlying structural heart diseases or arrhythmias (Class I; Level of Evidence C).

### 4.5. Very Long Pauses during Sleep in Athletes

When very long sinus pauses (e.g., >4 s) occur during sleep, even when associated with an abrupt increase in vagal tone, it may raise the concern of a propensity to develop pause-mediated polymorphic ventricular tachycardia or ventricular fibrillation. Although this has never been substantiated, this propensity could explain some unexpected sudden deaths in athletes during sleep. Such long pauses require discussion with the athlete about the potential for extreme remodeling. In some, these pauses have been associated with a decline in general performance (often interpreted as “overtraining syndrome”, a vaguely defined syndrome), which could hint at the need for temporary and/or partial training stoppage with follow-up on the reversibility of the sinus nodal function [30].

### 4.6. Cardioneuroablation in Athletes with SND

Proposed in the 1990s, cardioneuroablation is an endocardial atrial catheter ablation technique used to obtain enough vagal denervation to treat functional bradyarrhythmia without a pacemaker [31]. It refers to the modification of the efferent arm of the autonomic nervous system by radiofrequency catheter ablation of the main ganglionated plexi and has the potential to be a good treatment option for specific patients with vagally-mediated bradyarrhythmia [32]. This technique may reduce the impact of hypervagotonia on the heart [32]. Despite being a procedure that has shown great reproducibility and clear benefits to patients, its use requires great caution due to the complexity of cardiac innervation. The technique has a long and challenging learning curve, and it continues to be consistently improved by numerous investigators. Currently, there are no recommendations for this method for the treatment of athletes with SA or SND.

### 4.7. Shared Decision-Making in Cardiac Electrophysiology Procedures and Arrhythmia

Management shared decision-making (SDM) is a process through which patients and clinicians collaborate to make healthcare-related decisions while considering the available evidence, risk–benefit assessments, expected outcomes, and patient preferences and values [33]. SDM advances the ethical principle of patient autonomy. SDM has been advocated to improve patient care; decision acceptance; motivation; adherence; reported outcomes; preferences and values to make decisions; and patient-provider communication. SDM documentation is currently endorsed by several society guidelines and is a condition of reimbursement for selected cardiovascular and cardiac arrhythmia procedures. Therefore, shared decision-making in cardiac electrophysiology procedures and arrhythmia management is particularly important given the pros and cons of implementing or abandoning these procedures [34].

### 4.8. Is a Sports HRM Just a Gadget or a Reliable Medical Device?

Modern HRMs can be divided into their stripe (SHRMs) and optical (OHRMs) forms, both of which use different HR reading methods. SHRMs assess the main electric field produced during ventricle contraction, working similarly to ECGs. OHRMs use a phenomenon called photoplethysmography, which comprises illuminating the skin with a light source and measuring the amount of light that is scattered by blood flow [35].

Some HRMs with HRV function can also measure HR over a long period, allowing them to read the beat-to-beat distance (in ECG, it is the interval between the R-R waves of the QRS complexes) but not to determine whether the ventricular beat is conducted in the correct path or if there is an additional beat [36]. Furthermore, HRMs cannot recognize P-waves (which ECGs can), nor can they distinguish between supraventricular and ventricular tachycardia [37]. In summary, they enable the identification of the lack of ventricular activation but not the cause of this lack (e.g., whether it was an atrioventricular block or SA). Despite these shortcomings, modern sports HRMs are very effective non-medical devices for identifying tachyarrhythmias, especially those that are symptomatic [36]. They can accidentally identify bradyarrhythmia [38] and some can even make ECG recordings temporarily or during the entire training session [14].

Although past HRMs were seemingly non-significant for the treatment of asymptomatic exercise-stimulated arrhythmias [39], they became effective diagnostic tools for confirming the occurrence of symptomatic arrhythmias [36]. Indeed, research shows that modern HRMs (especially those with the ECG recording function) used by athletes of endurance disciplines are becoming a useful, important, and effective diagnostic tool in the detection and final diagnosis of cardiac arrhythmias (both symptomatic and asymptomatic) stimulated by exercise, having the potential to significantly contribute to safety during training [35]. HRMs can also be used in aquatic environments, thus being important for triathletes and long-distance swimmers, among other athletes [40]. They are also being increasingly used in cardiac rehabilitation [41] and follow-up during physical activity for patients with long QT syndrome [42,43]. Considering these advances, it can be assumed that future HRMs will have comparable diagnostic value for detecting cardiac arrhythmias as the Holter ECG has now. HRMs can surpass the current methods by enabling constant data transmission, being easier to use, and being more affordable.

The HRV function that some current HRMs have allows monitoring the heart for hours. This function helped to identify the pauses in the HR of the study participant. However, not all athletes realize that they have access to devices that are not only excellent training tools but can also serve as medical diagnostic equipment. The HRM featured in this article, the Polar 800V, requires its user to manually and intentionally activate the HRV function, and the activation of this function does not allow for simultaneously recording a training session on the HRM. However, it does allow for checking the number of hours of HRV (R-R distance) recording and provides a very detailed analysis. Figure 6 shows, in graphics, the recording shown in Figure 1, and illustrates the level of detail of the HR analyses delivered by some modern HRMs.

### 4.9. Limitations, Strengths, and Perspectives

The study has various limitations. First, there are problems regarding the definition of amateur sports (e.g., the amount of time that athletes spend training and their training intensity) and how they differentiate from professional sports. The literature on rhythm and conduction disturbances in the latter is extensive, whereas the effects of recreational activities on the sinus node are sparse [44,45].

Second, there were some problems regarding comparisons between HRM readings and Holter ECG readings; the process was difficult because assessing the reliability of the measurement of each R-R interval in “catching” the longest intervals was needed. Specifically, although the Holter ECG used in this study can analyze HRV and provide a summary of the results from the entire recording, it does not provide these values in the form of individual R-R intervals. This forced the investigators to evaluate individual R-R intervals in a semi-automated fashion, hindering the ability to make comparisons over many hours. Another significant technical problem was obtaining good quality recordings on both devices simultaneously (artifacts on one of them prevented comparative analysis).

Third is the uncertainty regarding the cause of the SA diagnosis. Although several clinical and diagnostic tests were performed to exclude other causes, the possibility of the SA having been caused by something other than the influence of exercise could not be completely eradicated. However, this pause mechanism was supported by the fact that the symptoms disappeared after the training stoppage. There was also a lack of follow-up for 6 years and of SA-related data thereof; during 4 of these years (2016–2020), the study participant practiced sports at a much lower load level, and for 2 years, the participant completely stopped training.

Fourth, it could not be guaranteed that the diagnosis of TIA was accurate. The normal MRI and CT brain imaging results and the ambiguous clinical picture (transient numbness of the upper limbs) suggest the potential for an unjustified diagnosis. However, the participant did, in fact, experience nocturnal HR pauses that went as high as 7.3 s, which are not normal and require diagnostic and therapeutic measures, although these should not be related to invasive methods.

We do not know what prompted the cardiologists after transferring the patient from the neurology department to consider such a decision. In the neurology department, the diagnosis of TIA was unequivocally made. According to some reports, the mere presence of SA is associated with a higher probability of strokes without an indicated (established) cause [46].

Researchers should conduct a large comparative study with samples of amateur athletes and inactive individuals in search of rhythm disturbances at the level of the sinus node. The data yielded by such research will be important for further developing our understanding of the mechanisms of bradyarrhythmia in athletes (e.g., whether it is related to the downregulation of HCN4 and funny current).

Despite these limitations, to the best of our knowledge, no prior case report study has described a situation wherein an amateur soccer player showed HR pauses that were pathologic and related to SA. Hence, this study reports on a unique, in-depth, long-term follow-up case of an amateur athlete with SA. It is possible that the athlete developed an abnormal “regulatory” response to physical activity, for which there is no clear evidence. An important fact is the absence of concomitant diseases in the described athlete, which could be responsible for bradyarrhythmia, unlike other publications [47].

This study also demonstrates the potential of sports HRMs with the HRV function, which are likely to be capable of capturing additional pauses during non-workout periods. Although sports HRMs are currently still generally designed to assess physical activity, they are also increasingly used as sources of medical knowledge, including cardiac bradyarrhythmia during rest periods [48].

## 5. Conclusions

Significant sinus arrest was observed in an amateur soccer player during the nighttime, which resolved after a period of detraining. The analyses do not allow for excluding this as being an abnormal cardiac response to physical activity. Proposing invasive therapy in such cases should be considered with caution (patient-centered care and shared decision-making in cardiac pacing therapy) after cessation of training. This case report showcases that some sports HRM with a HRV function can be important diagnostic tools for resting bradyarrhythmias, allowing for hours of observation while sleeping. Furthermore, this paper can shed new light on the “unconditional perceptions” of amateur physical activity. However, more cases and studies are needed to establish a potential constant relationship of life-threatening cardiac rhythm disorders and leisure time activity.

## Figures and Tables

**Figure 1 ijerph-19-10367-f001:**
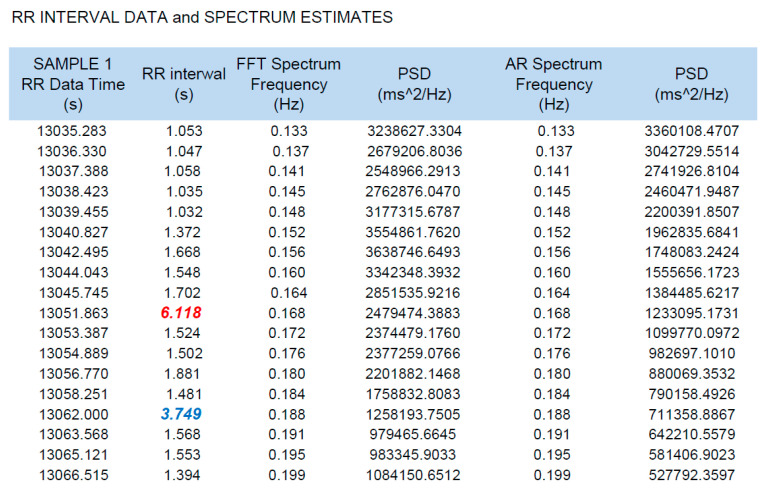
An example of a 30 s HRV printout recorded on an HRM that was compared with the ECG Holter recordings. The R-R interval (column 2) was evaluated while considering the recording time, here given in s after the start of the recording (column 1). The longest interval is 6.118 s, marked in red. The second longest interval is 3.749 s, marked in blue. R-R interval, interval between consecutive beats (i.e., the interval between two R-waves of QRS complexes in ECG); FFT, the fast Fourier transform; PSD, power spectral density; AR, autoregressive.

**Figure 2 ijerph-19-10367-f002:**
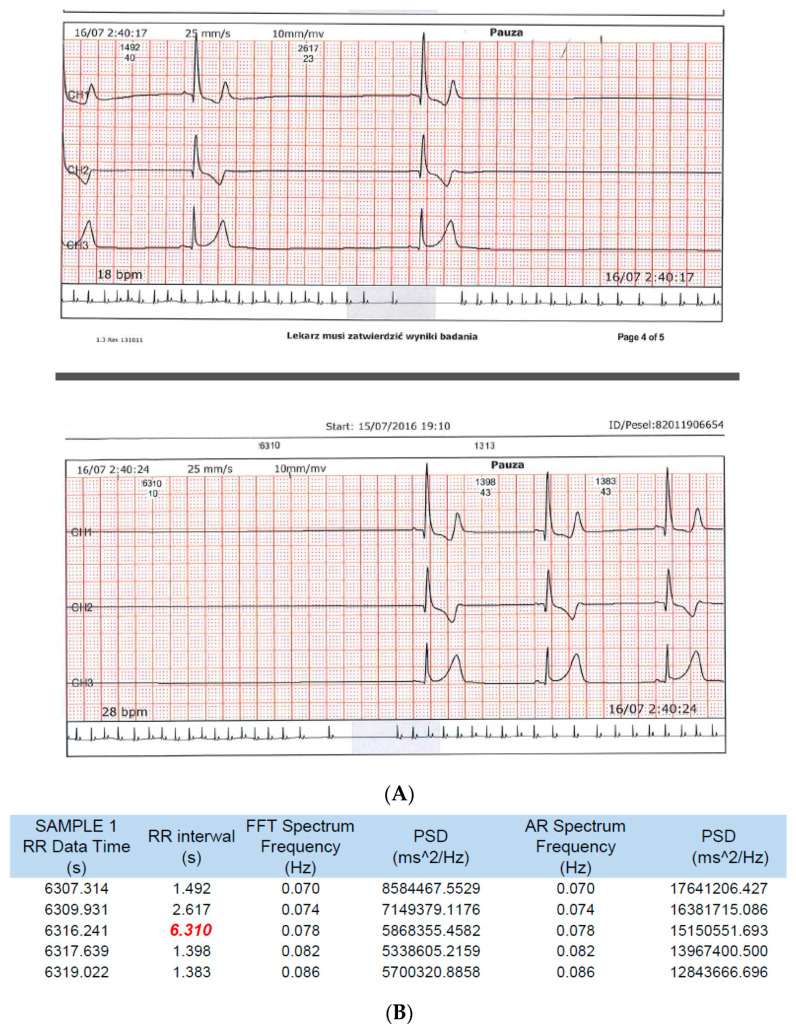
Holter ECG printout (**A**) and the corresponding printout from the HRV function of the HRM (**B**) showing the longest pause recorded on both devices. Red color-the longest pause.

**Figure 3 ijerph-19-10367-f003:**
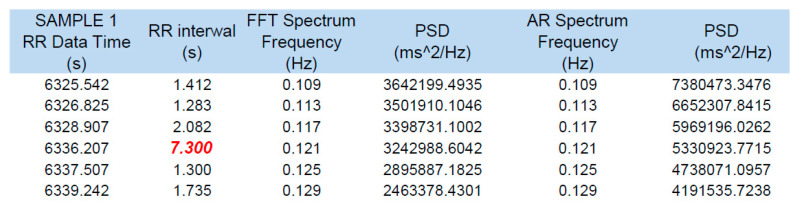
Longest pause recorded exclusively on the HRM through its HRV function, of 7.3 s. (red).

**Figure 4 ijerph-19-10367-f004:**
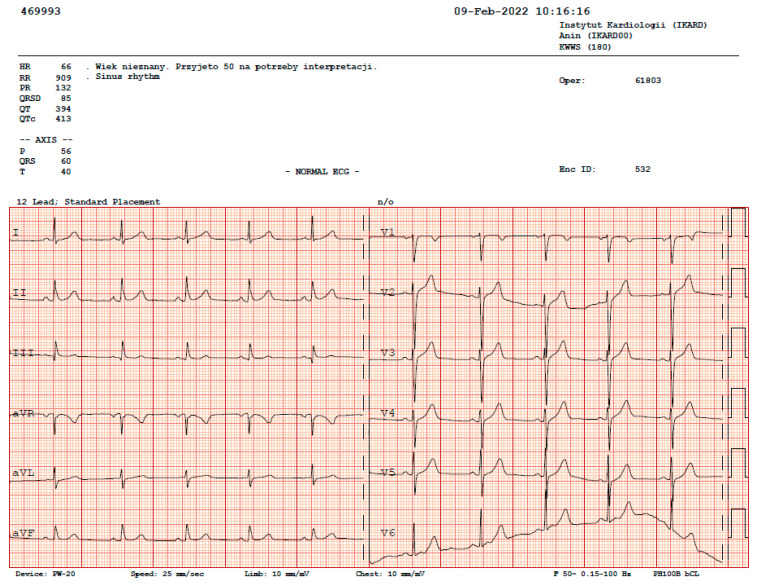
ECG of the examined athlete.

**Figure 5 ijerph-19-10367-f005:**
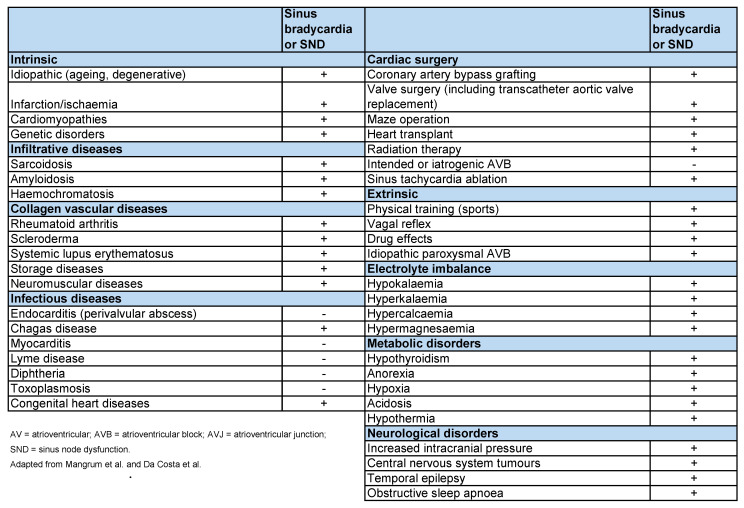
Intrinsic and extrinsic causes of bradycardia [16].

**Figure 6 ijerph-19-10367-f006:**
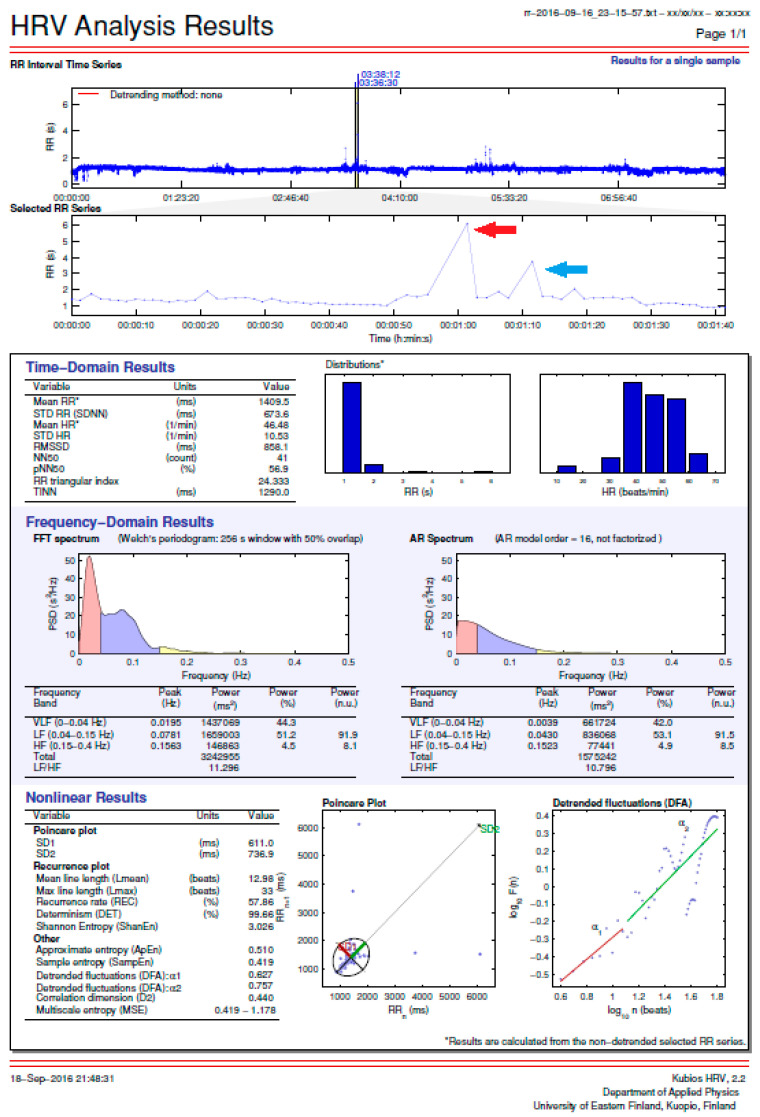
The HRV analysis capabilities of the Polar 800V HRM. The red arrow shows a pause of 6.118 s and the blue arrow shows a pause of 3.749 s between the R-R waves marked in the same color in Figure 1. The abbreviations in the figure are be defined because they are irrelevant to the purposes of this article.

**Table 1 ijerph-19-10367-t001:** Involvement in individual sports of the examined amateur athlete.

Age (Yrs)	Discipline	Training—Number of Hours/Week	Number of Years in Training	Hours of Training in the Sports Discipline	Comments:
9–20	Soccer	6	12	3700	Probably healthy
21–34	Indoor soccer	4	14	2900	Probably healthy
34–34	Break–2 months	0	0	0	Pauses on HRM
35–36	Taekwondo	6	2	600	No control
37–37	Break–2 months	0	0	0	No control
37–38	Taekwondo	6	4	1200	No control
39–40	Break–2 years	0	0	0	No control
Total years spent training in different sports disciplines: 32 years	All disciplines	About 5.5	32	8400 (about)	About 8400 h in total, on average 5.5 h/week/32 years

**Table 2 ijerph-19-10367-t002:** Summary analysis of the recordings from the Holter ECG and HRM.

Number	Hour	HR on Holter ECG					HR on HRM				
		Number of Recorded Beats	Mean (bpm) (s)	Min (bpm) (s)	Max (bpm) (s)	Pause > 4.0 s	Number of Recorded Beats	Mean (bpm) (s)	Min (bpm) (s)	Max (bpm) (s)	Pause > 4.0 s
1	02:00	2966	49.43	33.01	77.12	6.310	2960	49.33	32.98	77.08	6.310
	03:00		1.214	1.818	0.778			1.216	1.819	0.778	
2	00:00	3252	54.20	40.26	66.08	4.220	3268	54.47	40.27	66.04	4.202
	01:00		1.107	1.490	0.908			1.102	1.490	0.909	
3	01:00	3117	51.95	46.32	75.11	5.215	3111	51.85	46.35	75.13	5.211
	02:00		1.155	1.295	0.799			1.157	1.294	0.799	
4	05:00	3002	50.03	34.31	68.21	5.301	3010	50.17	34.29	68.19	5.310
	06:00		1.199	1.749	0.880			1.196	1.750	0.880	
5	04:00	3089	51.48	41.07	86.42	4.190	3091	51.52	41.09	86.39	4.199
	05:00		1.165	1.461	0.694			1.165	1.460	0.695	
6	00:00	3237	53.95	48.39	64.05	4.580	3242	54.03	48.41	64.04	4.589
	01:00		1.112	1.240	0.937			1.110	1.239	0.937	
7	05:00	3287	54.78	48.09	75.23	4.803	3284	54.73	48.07	75.27	4.805
	06:00		1.095	1.248	0.798			1.096	1.248	0.797	
8	02:00	3224	53.73	38.14	72.06	4.302	3215	53.58	38.16	72.09	4.309
	03:00		1.117	1.573	0.833			1.120	1.572	0.832	
9	02:00	3279	54.65	38.32	89.16	4.009	3284	54.73	38.29	89.17	4.002
	03:00		1.098	1.566	0.673			1.096	1.567	0.673	
10	03:00	3016	50.27	42.13	82.54	6.130	3020	50.33	42.16	82.52	6.135
	04:00		1.194	1.424	0.727			1.192	1.423	0.727	
	Total:	31,469				Total:	31,485				

Bpm, beat per minute; HR, heart rate; HRM, heart rate monitor.

**Table 3 ijerph-19-10367-t003:** Echocardiographic parameters of heart systolic and diastolic function.

Parameters	Units (Normal Values)	2016 Result 2022	
Left ventricular end-diastolic volume	mL (106 ± 22)	116	110
Left ventricular end-systolic volume	mL (41 ± 10)	36	35
Ejection fraction two-dimensional (%) bi-plane	% (62 ± 5)	65	65
Global longitudinal strain	% (−20)	20.6	20.8
Interventricular septum diameter	mm (6–10)	10	10
Posterior wall diastolic diameter	mm (6–10)	10	10
Right ventricular end-diastolic diameter	mm (20–30)	30	30
S’ right ventricle	cm/s (14.1 ± 2.3)	16	16
Left atrium	mm (30–40)	34	36
Left atrial volume index	mL/m^2^ (16–34)	28.0	28.4
Right atrial area	cm^2^ (16 ± 5)	14.0	14.0
Mitral valve E-wave	cm/s (73 ± 19)	78	80
Mitral valve A-wave	cm/s (69 ± 17)	50	52
E’ lateral	cm/s (>10)	20	18
E’ septal	cm/s (>7)	10	12
E/e’ lateral	ratio (<15)	4.0	4.2
E/e’ septal	ratio (<13)	7.0	7.2

**Table 4 ijerph-19-10367-t004:** Changes in recommendations for cardiac pacing and resynchronization therapy from 2013 to 2021.

	2013	2021
	Class	
Cardiac Pacing for Bradycardia and Conduction System Disease		
In patients with syncope, cardiac pacing may be considered to reduce recurrent syncope when asymptomatic pause(s) > 6 s due to sinus arrest are documented.	IIa	IIb

## Data Availability

Not applicable.

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
