# Peer review of "Amateur Athlete with Sinus Arrest and Severe Bradycardia Diagnosed through a Heart Rate Monitor: A Six-Year Observation—The Necessity of Shared Decision-Making in Heart Rhythm Therapy Management"

_ijerph, 2022, doi:10.3390/ijerph191610367_

Round 1

Reviewer 1 Report

Thank you for the chance of reviewing.

The authors showed a Case Report of sinus arrest observed in an amateur soccer player during the nighttime, which resolved after a period of retraining.

# I wonder if the patient may be asymptomatic during the clinical course. If the patients had no faintness or some symptom of bradycardia with normal heart function, we need not undergo PM imp for such an athlete heart patient.

# Sinus bradycardia has been considered as physiological changes secondary to long-term physical activity, a very low association with structural heart disease, and a low incidence of symptoms. Abandoning exercise or reducing the intensity or duration of physical activity might improve symptoms in those athletes with exercise-induced, symptomatic bradycardia. So, I seem strange in the presentation of this case.

# In addition, if the heart rate of the presented patient elevated after exercise at TMT, the HR response may be good enough to be free from the PM imp. Therefore, please show the results of HR from TMT in the presented patient.

# There is a similar article as follows. The patient was not followed for a long time as in the presented case. Please explain the novelty of this case.

Kaur S, Alsheikhtaha Z, Mehra R. A 48-Year-Old Athletic Man With Bradycardia

During Sleep. Chest. 2018 Nov;154(5):e139-e142.

Author Response

Dear Reviewer,

Thank you very much for your review of the paper. I tried to incorporate all your suggestions . If I have not succeeded, I will be happy to correct the work again.

With best regards

Author

PS: answers in the attached file. The work was linguistically corrected by a professional company ( certificate attached to the editor)

Reviewer 2 Report

Amateur Athlete with Sinus Arrest and Severe Bradycardia Diagnosed through a Heart Rate Monitor: A Six-year Observation. The Necessity of Shared Decision-making in Heart Rhythm Therapy Management

Comments and suggestions to authors

In the case report entitled: Amateur Athlete with Sinus Arrest and Severe Bradycardia Diagnosed through a Heart Rate Monitor: A Six-year Observation. The Necessity of Shared Decision-Making in Heart Rhythm Therapy Management, the authors reported a case of an amateur athlete with severe bradyarrhythmias, which resolved after deconditioning. It represents an interesting case, because although bradyarrhythmias are common in competitive athletes, there are no sufficient data regarding management of rhythm disorders in the amateur athletes. The authors also underline the usefulness of some contemporary sports heart rate monitors in detecting cardiac rhythm disorders and emphasize the value of shared decision-making. The authors, finally review the literature regarding SND in athletes, and eligibility criteria for competitive athletes with SND. I suggest the authors consider the following areas of improvement:

According to the ESC Guidelines on cardiac pacing and cardiac resynchronization therapy 2021: “In patients with syncope, cardiac pacing may be considered to reduce recurrent syncope when asymptomatic pauses>6s due to sinus arrest is documented. Establishing a correlation between symptoms and bradyarrhythmia is a crucial step in decision-making.” In this case, the athlete does not report any symptoms, since the bradyarrhythmias were an accidental finding during sleep. Also, the possible TIA (numbness) cannot be attributed to the cardiac rhythm disorders, in the absence of embolic foci (no tachycardia- bradycardia syndrome). You also express some skepticism in the limitation section. What is the proposed mechanism of the TIA that correlates it with SND in this patient?

Abstract:

“Prior to the visit to our center, the athlete was hospitalized and diagnosed with a transient ischemic attack, there he refused a pacemaker implantation.” According to the above, regarding the correlation of TIA and SND in this patient, I suggest the following: During the patient’s hospitalization for transient ischemic attack, the longest pauses on the Holter ECG were recorded, and the patient was suggested to undergo a pacemaker implantation.

“The significant SA resolved after a period of retraining.”: Perhaps you intended to write detraining instead of retraining.

Introduction:

“A single episode of transient ischemic attack (TIA), which manifested after the athlete woke up, served as a reason for a neurologist and a cardiologist to propose, in a joint consultation with the patient, a pacemaker implantation. The suspected cause of the TIA was SA.” I suggest that this part should be rephrased according to the above mentioned.

“emphasize the possibility of the presence of excessive bradyarrhythmias, often described in professional and amateur athletes, potentially shedding new light and problematizing the “unconditional perception” of physical activity as a form of healthy lifestyle.” More cases and studies are needed to establish a constant relationship of life-threatening cardiac rhythm disorders and leisure time activity. This is an isolated case, and no safe conclusion can be made for physical activity in general.

Discussion:

4.1 “This new exercise allowed him to perform physical activities without experiencing any symptoms suggestive of significant hemodynamic disturbances during sleep.”. Again, no symptoms were reported. I think that this sentence should be omitted.

4.3 “Meanwhile, SND can be defined by the presence of more than one of the following criteria: an average HR of <50beats/min at daytime; <40 beats/min at night; maximal R-R interval of at least 2.5 s; atrial flutter; and pacemaker implantation for SND [18]”. The next sentence in the manuscript includes the definition of SND according to the ESC guidelines 2021, so this is unnecessary and confusing.

Conclusion:

“Significant sinus arrest was observed in an amateur soccer player during the nighttime, which resolved after a period of retraining.” As in the abstract session, I think you meant detraining, instead of retraining.

“Furthermore, this paper sheds new light on the "unconditional perceptions" of amateur physical activity as a form of healthy lifestyle.” I think that this conclusion should be toned down, as explained above.  

Moderate English changes required.

Reference 36 is by mistake mentioned as 34.

Author Response

Dear Reviewer,

Thank you very much for your review of the paper. I tried to incorporate all your suggestions . If I have not succeeded, I will be happy to correct the work again.

With best regards

Author

PS: answers in the attached file. The work was linguistically corrected by a professional company ( certificate attached for the editor)

Round 2

Reviewer 1 Report

Now, I have no more question.

Author Response

Dear Reviewer 1

Thank you again for your review of the article and its acceptance in  current form.

Author.

Reviewer 2 Report

Amateur Athlete with Sinus Arrest and Severe Bradycardia Diagnosed through a Heart Rate Monitor: A Six-year Observation. The Necessity of Shared Decision-making in Heart Rhythm Therapy Management

The comments have been addressed adequately.

Minor comment in the introduction section: "emphasize the potential possibility of the presence of excessive bradyarrhythmias, often described in professional and amateur athletes, potentially shedding new light and problematizing the unconditional perception of physical activity as a form of healthy lifestyle." I think the phrase potential possibility is an exaggeration..

There is indeed an overlap with the methodology section of an article published by the same authors (https://doi.org/10.3390/diagnostics10060391). The similarity can be attributed to the fact, that both cases concern rhythm disorders in athletes. So, they have the same rationale and similar tests are required to investigate both cases. However, the authors could perhaps rephrase the methodology section of the article under review to avoid the overlap.

Author Response

Response to Reviewer 2

Thank you for your review of the article, as well as important tips, questions and critical comments, the inclusion of which will definitely improve the value of the article.

Remark:

The comments have been addressed adequately.

Minor comment in the introduction section: "emphasize the potential possibility of the presence of excessive bradyarrhythmias, often described in professional and amateur athletes, potentially shedding new light and problematizing the “unconditional perception” of physical activity as a form of healthy lifestyle." I think the phrase potential possibility is an exaggeration..

Answer:

I have removed this phrase and changed the aim 2

Before the change

  1. emphasize the potential possibility of the presence of excessive bradyarrhythmias, often described in professional and amateur athletes, potentially shedding new light and problematizing the “unconditional perception” of physical activity as a form of healthy lifestyle. (However, more cases and studies are needed to establish a potential constant relationship of life-threatening cardiac rhythm disorders and leisure time activity. This is an isolated case, and no safe conclusion can be made for physical activity in general).

After the change

  1. draw attention to the fact of excessive bradyarrhythmias in the amateur athlete, described mainly in professional endurance athletes.

Remark:

There is indeed an overlap with the methodology section of an article published by the same authors (https://doi.org/10.3390/diagnostics10060391). The similarity can be attributed to the fact, that both cases concern rhythm disorders in athletes. So, they have the same rationale and similar tests are required to investigate both cases. However, the authors could perhaps rephrase the methodology section of the article under review to avoid the overlap.

Answer:

You are undoubtedly correct that both cases involve rhythm disorders in athletes as well as that similar tests are required to investigate both cases. A reformulation of the methodology would indeed violate the benchmarks of the diagnostic laboratories in which the tests were performed. The results are different , the methodology is the same.

I would like to ask you to accept the current state.

Best regards

Author
